# *cis*-Decalin-containing tetramic acids as inhibitors of insect steroidogenic glutathione *S*-transferase Noppera-bo

**Naoki Kato**[1,2☯]*, **Kana Ebihara**[3☯], **Toshihiko Nogawa**[4,5], **Yushi Futamura**[4,6], **Kazue Inaba**[3], **Akiko Okano**[4,6], **Harumi Aono**[4,6], **Yuuta Fujikawa**[7], **Hideshi Inoue**[7], **Kazuhiko Matsuda**[8,9], **Hiroyuki Osada**[4,6], **Ryusuke Niwa**[10], **Shunji Takahashi**[1]

1 Natural Product Biosynthesis Research Unit, RIKEN Center for Sustainable Research Science, Wako, Saitama, Japan, 2 Faculty of Agriculture, Setsunan University, Hirakata, Osaka, Japan, 3 Degree Programs in Life and Earth Sciences, Graduate School of Science and Technology, University of Tsukuba, Tsukuba, Ibaraki, Japan, 4 Chemical Biology Research Group, RIKEN Center for Sustainable Research Science, Wako, Saitama, Japan, 5 Molecular Structure Characterization Unit, RIKEN Center for Sustainable Research Science, Wako, Saitama, Japan, 6 Chemical Resource Development Research Unit, RIKEN Center for Sustainable Research Science, Wako, Saitama, Japan, 7 School of Life Sciences, Tokyo University of Pharmacy and Life Sciences, Hachioji, Tokyo, Japan, 8 Department of Applied Biological Chemistry, Faculty of Agriculture, Kindai University, Nara, Nara, Japan, 9 Agricultural Technology and Innovation Research Institute, Kindai University, Nara, Nara, Japan, 10 Life Science Center for Survival Dynamics, Tsukuba Advanced Research Alliance (TARA), University of Tsukuba, Tsukuba, Ibaraki, Japan

☯ These authors contributed equally to this work.
* naoki.kato@setsunan.ac.jp

**Data Availability Statement:** All relevant data are within the paper and its Supporting information files.

## Abstract

Decalin-containing tetramic acid is a bioactive scaffold primarily produced by filamentous fungi. The structural diversity of this group of compounds is generated by characteristic enzymes of fungal biosynthetic pathways, including polyketide synthase/nonribosomal peptide synthetase hybrid enzymes and decalin synthase, which are responsible for the construction of a linear polyenoyl tetramic acid structure and stereoselective decalin formation via the intramolecular Diels–Alder reaction, respectively. Compounds that differed only in the decalin configuration were collected from genetically engineered mutants derived from decalin-containing tetramic acid-producing fungi and used for a structure-activity relationship study. Our evaluation of biological activities, such as cytotoxicity against several cancer cell lines and antibacterial, antifungal, antimalarial, and mitochondrial inhibitory activities, demonstrated that the activity for each assay varies depending on the decalin configurations. In addition to these known biological activities, we revealed that the compounds showed inhibitory activity against the insect steroidogenic glutathione *S*-transferase Noppera-bo. Engineering the decalin configurations would be useful not only to find derivatives with better biological activities but also to discover overlooked biological activities.

**Funding:** This work was supported by JSPS KAKENHI (Grant numbers 18K19163 to RN, 19H04665 and 20K05872 to NK, and 21H04718 to RN, NK, and KM) and the Support for Pioneering Research Initiated by the Next Generation, Japan Science and Technology Agency (JST SPRING; Grant number JPMJSP2124). KE received a fellowship from JST SPRING. The funders had no role in study design, data collection and analysis, decision to publish, or preparation of the manuscript.

**Competing interests:** The authors have declared that no competing interests exist.

## Introduction

Decalin is a bioactive scaffold found in natural products primarily produced by microorganisms, including filamentous fungi and actinomycetes [1, 2]. The bicyclic structure is highly substituted with various functional groups and connected to other cyclic structure types, including lactone, pyrone, tetramic acid (pyrrolidin-2-one), and pyrrolizidine, which generates diverse and complex chemical structures. This group of compounds has been reported to exhibit various biological activities, including antibacterial, antifungal, antiviral, and antihyperlipidemic activities [1]. Decalin-containing tetramic acid produced by filamentous fungi, represented by HIV-1 integrase inhibitor equisetin (**1**) [3] and its stereochemical opposite, phomasetin (**2**) [4], is one such group, and is biosynthesized by polyketide synthase (PKS)-nonribosomal peptide synthetase (NRPS) hybrid enzyme pathways [2, 5]. The pathways include multiple processes for structural diversification: an iterative PKS module of the PKS-NRPS hybrid enzyme, in collaboration with trans-acting enoylreductase, controls the number of polyketide extension cycles, methylation patterns and their stereochemistry; a single-acting NRPS module determines an amino acid substrate, constituting the tetramic acid moiety; and decalin synthase (DS) [6–8] catalyzes stereoselective decalin formation from a linear polyenoyl tetramic acid via an intramolecular Diels–Alder reaction (S1 Fig).

Structure-activity relationship (SAR) studies of decalin-containing tetramic acids using their derivatives and related compounds have suggested the importance of the tetramic acid moiety in exerting biological activities [2]; however, little is known regarding the relationship between biological activity and the decalin moiety, particularly the decalin configuration. Because *trans-* and *cis-*decalins have been considered different carbon skeletons, the simultaneous synthesis of both *trans-* and *cis-*decalin compounds and their biological evaluation are challenging. Although normal biosynthetic pathways involving DS-mediated cyclization yield biosynthetic products with a single decalin configuration under standard culture conditions, genetically engineered mutants lacking DS genes produce a mixture of diastereomers with different decalin configurations [6, 7, 9]. For example, the deletion mutant of the DS gene *fsa2*, derived from a **1**-producer fungus *Fusarium* sp. FN080326 [10], produced **1** (2*S*,3*R*,8*S*,11*R*) and its *cis-*decalin diastereomer (2*S*,3*R*,8*R*,11*S*) [6]. Amino acid substitutions in CghA, an Fsa2 homolog involved in Sch 210972 biosynthesis, altered the reaction selectivity to form *cis-*decalin with a 2*R*,3*S*,8*S*,11*R* configuration instead of the natural *trans-*decalin (2*R*,3*S*,8*R*,11*S*) [11]. The genetic and protein engineering of the key enzyme responsible for the stereoselective Diels–Alder reaction could provide a means of creating derivatives with different decalin configurations.

In this study, we examine the biological activities of decalin-containing tetramic acids collected from the **1**-producer fungus, *Fusarium* sp. FN080326, and its *fsa2* deletion mutant [6], and a phomasetin (**2**)-producer fungus, *Pyrenochaetopsis* sp. RK10-F058, and its mutants lacking *phm5* and *phm7*, and replacing *phm7* with *fsa2* and mutated *phm7* [12, 13], providing a SAR, particularly between decalin configurations and biological activities, including cytotoxicity, antimicrobial activity, mitochondrial inhibition, and the insect steroidogenic glutathione *S*-transferase (GST), Noppera-bo.

## Materials and methods

### Chemical analysis

All solvents and reagents were of analytical grade and were obtained from commercial sources. The UV spectra, optical rotations, and electronic circular dichroism (ECD) spectra were recorded using a Beckman DU 530 Life Science UV/Vis spectrophotometer (Beckman

Coulter, Brea, CA, USA), HORIBA SEPA-300 high sensitive polarimeter (HORIBA, Kyoto, Japan), and JASCO J-720 spectropolarimeter (JASCO, Tokyo, Japan), respectively. Infrared spectra were recorded using a HORIBA FT-720 IR spectrometer equipped with a DuraSampl IR II ATR instrument. NMR spectra were recorded using a JEOL ECA-500 FT-NMR spectrometer at 500 MHz for $^1$H NMR and at 125 MHz for $^{13}$C NMR (JEOL, Tokyo, Japan). Chemical shifts were reported in ppm and referenced to the residual chloroform signals ($\delta_H$ 7.24 and $\delta_C$ 77.23 ppm). High-resolution electrospray ionization time-of-flight mass spectrometry (HRESITOFMS) was performed using a Waters Synapt GII (Waters, Milford, MA, USA). Medium-pressure liquid chromatography was performed using a Teledyne ISCO CombiFlash Companion (Teredyne ISCO, Lincoln, NE, USA). Preparative HPLC was performed using a Waters 600E pump system equipped with a Cosmosil C18 MS-II or AR-II column (Nacalai Tesque, Kyoto, Japan).

## Preparation of compounds 1–7

Compounds **1**–**7** tested in this study were prepared as described previously [6, 12, 13].

## Isolation and structure determination of compound 8

The Δ*phm7* mutant derived from the **2**-producer fungus *Pyrenochaetopsis* sp. RK10-F058 [13] was cultured at 28 ºC for 16 d in YMGS medium. Culture broth (10 L) was extracted three times with half the volume of EtOAc. It was evaporated to yield 1.23 g of a crude extract and then subjected to $SiO_2$-MPLC with a $CHCl_3$/MeOH stepwise gradient to obtain 8 fractions. The 5th MPLC-fraction eluted with $CHCl_3$/MeOH (90:10) was separated by $SiO_2$-MPLC with a hexane/EtOAC linear gradient to afford 7 subfractions. A powder-like insoluble material precipitated from the 2nd subfraction. It was suspended in MeOH, and the soluble material was removed to afford 46.7 mg of **8** as a white powder.

Its molecular formula was determined as $C_{24}H_{31}NO_6$ using HRESITOFMS. It was not soluble in any solvent, such as hexane, $CHCl_3$, EtOAc, acetone, MeOH, MeCN, DMSO, or $H_2O$, at sufficient concentrations to obtain physicochemical properties, including optical rotation, UV, and NMR spectra, except for IR, which was measured in the solid state and implied the presence of a hydroxyl group (3318 cm$^{-1}$). Therefore, methylation by trimethylsilyldiazomethane was applied and afforded a dimethylester of **8**, which was confirmed by the increased mass of 28 in $m/z$, which improved its solubility in $CHCl_3$ and MeOH.

The structure was investigated using the dimethylester. The $^1$H and $^{13}$C NMR spectra in $CDCl_3$ showed doubling signals with the ratio of approximately (3:2) for major and minor ones (S2 and S3 Figs). This is a characteristic feature of compounds containing a decalin skeleton connected to a tetramic acid, such as equisetin [14] and phomasetin [4]. The structure was determined by focusing on the major signals. The $^1$H NMR spectrum showed 4 methyl signals, including 1 doublet and 3 singlets, one of which was supposed to be attached to sp$^2$ carbon, as implied by the chemical shift observed at 1.58 ppm. The remaining 2 singlet methyls were observed at 3.72 and 3.76 ppm and assigned to the methoxy groups. However, no signal was observed for the NMe group, which is typical for phomasetins. Instead of the lack of an NMe signal, an exchangeable signal was observed at 5.89 ppm as a broad singlet, which was supposed to be the NH signal at the 1′ position. The $^{13}$C NMR spectrum showed 24 signals as major ones, including 4 carbonyl or related signals, one of which was observed at 206.4 ppm that implied the presence of a ketone, 7 sp$^2$ signals, and 4 methyl signals, two of which were assigned as methoxy groups based on their chemical shifts at 51.7 and 61.3 ppm. The direct connection between proton and carbon was confirmed by the correlation observed in HSQC spectrum with the support of the $^{13}$C DEPT 135 experiment (S4 and S5 Figs). Partial

structures, C-3 to C-9 and between C-3 and C-8, C-11 to C-16, and NH to C-6′, were constructed using correlations in the DQF-COSY and HSQC-TOCSY spectra (S6 and S7 Figs). Partial structures were constructed using the HMBC long-range correlations (S8 Fig). In particular, the terminal methyl ester was confirmed by HMBC correlations from 17-OMe to C-17 and from H-15 and H-16 to C-17. The HMBC correlation from another OMe to C-4′ allowed to assign the attachment of the OMe group at C-4′. Therefore, the planar structure of the dimethylester of **8** was constructed. The minor isomer was supposed to have the same planar structure as the major isomer by comparing their NMR chemical shifts. The isomer was presumably a rotamer at the C-C bond between C-1 and C-3' by the relatively large differences of $^{13}$C NMR chemical shifts around the bond (S1 Table).

The geometries of $\Delta^{13}$ and $\Delta^{15}$ of the side chain at C-3 were determined as *E* by the large coupling constants of 14.9 and 15.4 Hz in the $^1$H NMR spectrum. The relative stereostructure of the decalin skeleton was determined using NOESY correlations (S9 Fig). A *cis* configuration of the decalin skeleton was confirmed by the NOESY correlation between H-3 and H-8. H-3 also correlated to H-5 at 0.84 ppm and H-7 at 1.17 ppm, suggesting that they were axials and a chair conformation. The correlations between H-13 and both H-4ax at 1.29 ppm and Me-12 allowed to conclude that they were on the same side, and Me-12 and the side chain at C-3 were on the α-side. Me-19 was assigned as β by the correlation between H-6 and H-9.

The absolute structure on the decalin moiety was deduced as identical to those of **4** by the same negative specific rotation (**8**: $[\alpha]_D^{25}$−270˚ [*c* 0.1, MeOH] and **4**: $[\alpha]_D^{25}$−135˚ [*c* 0.05, MeOH]) and similar cotton effects of ECD spectra (**8**: positive to negative, then app. 0 at 224, 270, and 290 nm, **4**: positive to negative, then positive at 200, 235, and 285 nm). The hydroxymethyl group at C-5′ was implied as *R* by the biosynthesis background. These absolute configurations were confirmed by comparing the experimental ECD spectra with the calculated spectra of the (2*R*,3*S*,6*S*,8*S*,11*R*,5′*R*) and (2*S*,3*R*,6*R*,8*R*,11*S*,5′*S*) isomers. The resulting ECD spectrum of the (2*R*,3*S*,6*S*,8*S*,11*R*,5′*R*) isomer showed good agreement with the experimental ECD spectrum and contrasted with the opposite one, suggesting that the dimethyl ester of **8** had a (2*R*,3*S*,6*S*,8*S*,11*R*,5′*R*) configuration (S10 Fig). Based on these results, the structure of the dimethylester of **8** including its absolute configuration was determined. Therefore, **8** was determined to be the carboxylic acid derivative at the C-16 position of **4** (S11 Fig).

**Compound 8**: white powder; UV (MeCN/H$_2$O + 0.05% HCOOH on DAD-LC-MS analysis) $\lambda_{max}$ 266 nm; IR $\nu_{max}$ (ATR) 3318, 3191, 2911, 1683, 1637, 1556, 1455, 1373, 1243, 1193, 1114, 1004, 902, 763 cm$^{-1}$; HRESITOFMS found *m/z* 430.2226 [M+H]$^+$ calculated for C$_{24}$H$_{32}$NO$_6$ 430.2230.

## Preparation of the dimethylester of 8

To stirred solution of **8** (10.3 mg, 24.0 μmol) in benzene-MeOH (4:1, 10 mL) was added trimethylsilyldiazomethane (2.0 M hexane solution, 43.2 μL, 86.4 μmol). The mixture was stirred at room temperature for 4 h and concentrated. The residue was purified by SiO$_2$-MPLC and C18-HPLC to afford 2.0 mg of dimethylester of **8** as a colorless solid: $[\alpha]_D^{25}$−270˚ (*c* 0.1, MeOH); UV (MeOH) $\lambda_{max}$ (log ε) 266 (4.57) nm; $^1$H and $^{13}$C NMR chemical shifts are summarized in S1 Table.

## Computer calculation

Conformational analysis and optimization of conformers were performed using Spartan'20 [15], and each optimized conformer was used to simulate the ECD spectrum using Gaussian 16 C0.1 [16]. The simulated ECD spectra for each conformer were averaged based on the

Boltzmann average to obtain the calculated ECD spectra (S12 and S13 Figs). The calculations were performed using the same procedure as previously reported [17].

### *In vitro* cytotoxicity and antimicrobial assays

*In vitro* cytotoxicity and antimicrobial assays were conducted as previously reported [18, 19].

### Measurement of mitochondrial respiration

Changes in oxygen consumption rate (OCR) and extracellular acidification rate (ECAR) were measured using a Seahorse XFe96 analyzer (Agilent, CA, USA). The methods for measurement and analysis have been described in a previous report [20].

### Nobo activity inhibition assays

Recombinant Nobo proteins from the fruit fly *Drosophila melanogaster* (DmNobo) and the yellow fever mosquito *Aedes aegypti* (AeNobo) were prepared as previously described [21, 22]. The concentration of 50% inhibition ($IC_{50}$) was measured using an *in vitro* assay system as described previously [21, 22]. Briefly, the enzymatic activity of DmNobo and AeNobo was detected using 3,4-dinitrobenzamidedichlorofluorescein (3,4-DNADCF) [23]. In this study, a dilution series of decalin compounds from 20 µM to 39 nM was prepared by 2-fold serial dilution with dimethyl sulfoxide.

## Results

### Decalin-containing tetramic acids collected from genetically engineered fungi

Compound **1** containing a *trans*-decalin with the 2*S*,3*R*,8*S*,11*R* configuration and its *cis*-decalin isomer **3** (2*S*,3*R*,8*R*,11*S*) were obtained from the culture broth of the producer strain *Fusarium* sp. FN080326 and the decalin synthase *fsa2* deletion mutant [6, 10]. The other pair of *trans* and *cis*-decalins (2*R*,3*S*,8*R*,11*S* and 2*R*,3*S*,8*S*,11*R*), *N*-demethylphomasetin, **5**, and its *cis*-decalin isomer **4**, were collected from deletion mutants of *N*-methyltransferase *phm5* and decalin synthase *phm7*, respectively, derived from the **2**-producer fungus *Pyrenochaetopsis* sp. RK10-F058 [13, 24]. Compounds **1** and its *cis*-decalin derivative **3** were enantiomerically opposite to **2** and **5**, and **4**, respectively, although some differences were present in the substituents attached to the decalins (Fig 1); thus, decalin-containing tetramic acids with four possible decalin configurations were efficiently collected using producer fungi and their genetically modified mutants. In addition, the gene replacement mutants of *phm7* with *fsa2* and *phm7* containing the W342A mutation produced a **2** derivative with the unnatural **1**-type decalin (**6**) [13] and 17-hydroxy derivative **7** [12], respectively. Further investigation of the culture broth of the Δ*phm7* mutant allowed us to isolate an oxidized derivative of **4** (**8**).

Compound **8** was obtained as a white powder, and its molecular formula was determined as $C_{24}H_{31}NO_6$ using HRESITOFMS. The solubility of **8** in any solvent was too low to obtain its physicochemical properties. Therefore, it was methylated by trimethylsilyldiazomethane, affording a dimethylester of **8**, which was confirmed by the increased mass of 28 in *m/z* and improved solubility in $CHCl_3$ and MeOH. The structure of the dimethylester was determined by a combination of spectroscopic methods, including NMR and mass spectrometry, and the calculation of the ECD spectra (S2–S13 Figs, and S1 Table). Compound **8** was a carboxylic acid derivative at C-16 of the *cis*-decalin **4** (Fig 1).

**Fig 1. Chemical structures of decalin-containing tetramic acids 1–8.**

## Decalin configuration impacts biological activities

The metabolites collected from genetically engineered fungi share a decalin scaffold connected to a tetramic acid via a carbonyl group with a range of substituents attached to decalin (Fig 1) and thus are useful for SAR studies of their biological activities. Compounds **1**–**8** were first evaluated for their cytotoxicity against several cancer cell lines and their antimicrobial activities against bacteria, pathogenic fungi, and malarial parasites (Table 1).

Most compounds showed moderate cytotoxicity in the tested cancer cell lines. There was no significant difference in the cytotoxicity between *trans* decalin compounds **1**, **2**, and **5** and *cis* decalin compounds **3** and **4**, although **4** presented no cytotoxicity against HeLa and *src*^ts^-NRK cells. In contrast, **1** presented strong growth inhibitory activity against the Gram-positive bacterium *S. aureus* and plant pathogenic fungus *P. oryzae*, with $IC_{50}$ values of 1.4 and 0.15 μM, respectively, whereas **3** showed 4- and 3-fold lower activity compared with that of **1**. Similarly, the potent antimicrobial activities of **2** and its *N*-demethyl analog **5** against *S. aureus* and *P. oryzae* were reduced when decalin had a *cis*-configuration (i.e., **4**). Compound **6**, with an unnatural *trans*-decalin configuration, showed the lowest cytotoxicity and antimicrobial activity among the compounds tested. These results suggested that the decalin configuration affected the cytotoxicity and antimicrobial activity, and the *trans* configuration was preferable to the activity, particularly to the anti-*Staphylococcus* activity. Additionally, **7** and **8**, which had hydroxylated and carboxylated olefins attached to the C11 of the *trans*- and *cis*-decalin

**Table 1. *In vitro* cytotoxicity and antimicrobial activities of compounds 1–8.**

| Compound | Decalin configuration | Cytotoxicity (IC$_{50}$, µM) | | | Antimicrobial activity (IC$_{50}$, µM) | | | | | |
|---|---|---|---|---|---|---|---|---|---|---|
| | | HeLa | HL-60 | *src*$^{ts}$-NRK | *Sa* | *Ec* | *Af* | *Po* | *Ca* | *Pf* |
| 1 | 2*S*,3*R*,8*S*,11*R* | 35 | 12 | 20 | 1.4 | 13 | 29 | 0.15 | 29 | 12 |
| 6 | 2*S*,3*R*,8*S*,11*R* | >70 | 12 | >70 | 3.8 | >70 | >70 | 5.5 | 65 | 23 |
| 3 | 2*S*,3*R*,8*R*,11*S* | 46 | 14 | 35 | 5.6 | 20 | 40 | 0.46 | 40 | 13 |
| 2 | 2*R*,3*S*,8*R*,11*S* | 6.1 | 3.4 | 11 | 0.14 | 1.3 | 21 | 0.044 | 21 | 1.8 |
| 5 | 2*R*,3*S*,8*R*,11*S* | 40 | 11 | 63 | 0.20 | 2.5 | 9.8 | 1.3 | 3.8 | 4.8 |
| 7 | 2*R*,3*S*,8*R*,11*S* | 28 | 22 | n.t.[a] | 1.4 | 21 | >70 | 12 | >70 | 4.7 |
| 4 | 2*R*,3*S*,8*S*,11*R* | >75 | 22 | >75 | 6.0 | 1.5 | 58 | 2.0 | 9.0 | 13 |
| 8 | 2*R*,3*S*,8*S*,11*R* | >70 | >70 | >70 | 63 | 28 | >70 | >70 | >70 | 26 |

Notes: HeLa, human cervical epidermoid carcinoma cell line; HL-60, human promyelocytic leukemia cell line; *src*$^{ts}$-NRK, normal rat kidney cells infected with ts25, a T-class mutant of the *Rous sarcoma* virus Prague strain; *Sa*, *Staphylococcus aureus* 209; *Ec*, *Escherichia coli* HO141; *Af*, *Aspergillus fumigatus* Af293; *Po*, *Pyricularia oryzae* kita-1; *Ca*, *Candida albicans* JCM1542; *Pf*, *Plasmodium falciparum* 3D7

[a]n.t., not tested.

skeletons, respectively, caused a significant loss of biological activity, suggesting that the olefin moiety plays a role in exerting the activity.

In addition to its cytotoxicity and antimicrobial activities, **1** has been reported to inhibit mitochondrial respiratory chain and subsequent oxidative phosphorylation (OXPHOS) indirectly by influencing substrate anion carriers of mitochondria in eukaryotic cells [25]. To confirm this, the effect of **1** on energy metabolism was evaluated using an extracellular flux analyzer, which can simultaneously measure two metabolic properties: oxygen consumption rate (OCR) for an indicative of OXPHOS and extracellular acidification rate (ECAR) for that of glycolysis [20]. The data showed that **1** caused a decrease in OCR and a concomitant increase in ECAR, confirming its inhibitory activity of mitochondrial respiration (Fig 2, and S14 Fig). Then, we investigated the relationship between decalin configuration and mitochondrial inhibitory activity. Among the compounds tested (**2–5** and **8**), *cis*-decalin **3** showed a phenotype similar to **1** (Fig 2, and S14 Fig). Notably, in contrast to the antibacterial activity, *cis*-decalin **3**, but not **2** or **5**, had higher mitochondrial inhibitory activity than the *trans* isomer **1**, indicating that decalins with 3*R* configuration tend to inhibit substrate anion carriers.

### Decalin-containing tetramic acids inhibit the ecdysteroidogenic glutathione *S*-transferase Noppera-bo from the fruit fly *Drosophila melanogaster*

Changes in decalin configuration caused a decrease in antimicrobial activity and cytotoxicity and an increase in mitochondrial inhibitory activity. Considering that the stereochemistry of decalin compounds was directly linked to their biological activities, the compounds with four possible decalin configurations could be useful to discover previously overlooked biological activities. We postulated that the decalin-containing tetramic acids may have activity against insect steroidogenesis, in addition to the aforementioned biological activities, according to the following reasons. (i) Carbon skeleton of *cis*-decalin is structurally similar to that of the AB-ring moiety of ecdysteroids. (ii) Lovastatin and mevastatin with a decalin skeleton have been reported to inhibit the HMG-CoA reductase, a key enzyme in the steroid biosynthesis [26, 27]. (iii) the mammalian female hormone 17β-estradiol inhibits Noppera-bo (Nobo), a GST involved in the insect steroidogenesis [22]. Hence, to test this hypothesis, we investigated the

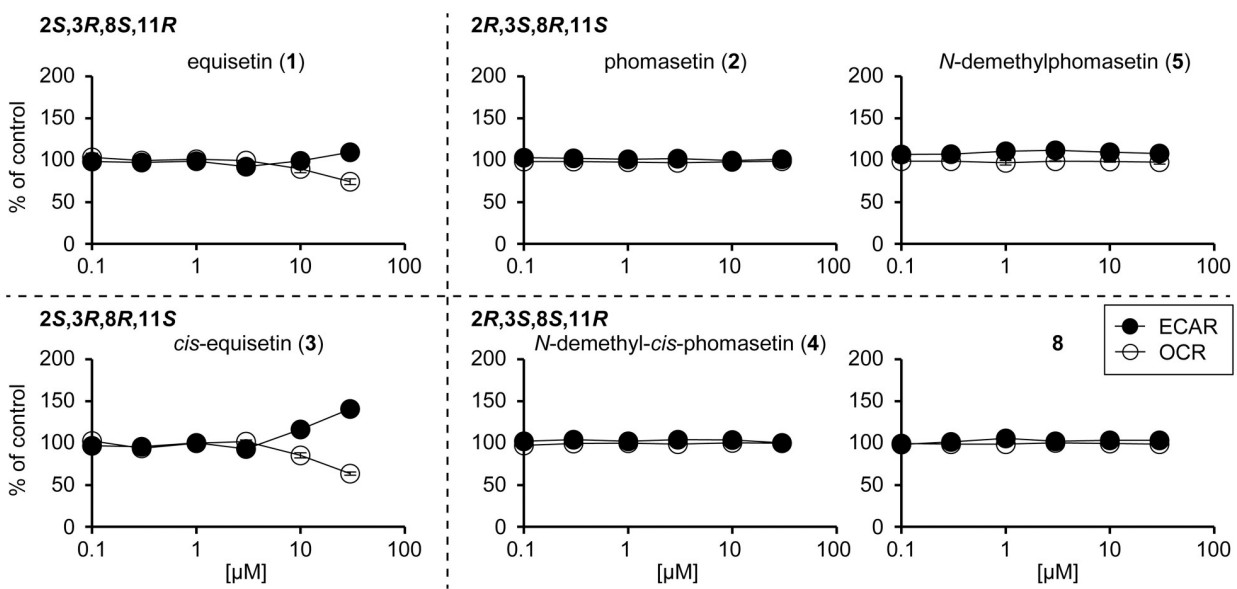

**Fig 2. Changes in cellular metabolism induced by decalin-containing tetramic acids.** Percentage changes in OCR and ECAR (relative to their respective baseline values) induced by test compounds were calculated from S14 Fig. Each data point represents the mean ± standard deviation value at 6 min after sample treatment (n = 3). Note that decreasing OCR and increasing ECAR is a typical phenotype characteristic to mitochondrial respiration inhibitors [20].

inhibitory activity of this group of compounds against Nobo. The inhibitory activity of compounds **1**–**5** was examined using an *in vitro* enzyme assay system with a fluorogenic artificial substrate, 3,4-DNADCF [23], and recombinant Nobo proteins from the fruit fly *Drosophila melanogaster* (DmNobo) or the yellow fever mosquito *Aedes aegypti* (AeNobo). In this assay system, the GSTs catalyze glutathione (GSH) conjugation to the weakly fluorescent 3,4-DNADCF, giving rise to a highly fluorescent product [21, 22, 23].

*cis*-decalins **3** and **4** exhibited inhibitory activity against DmNobo with $IC_{50}$ values of 8.04 ± 2.72 and 9.04 ± 3.32 μM, respectively (Fig 3). *trans*-decalins **1**, **2**, and **5** also showed moderate activity, whereas mevastatin, another representative fungal decalin compound that inhibits HMG-CoA reductase [26, 27], had little effect on DmNobo activity (S15 Fig). No tested compounds showed any inhibitory activity against AeNobo (Fig 3). Similar to the mitochondrial inhibitory activity, *cis*-decalin compounds showed higher DmNobo inhibitory activity than *trans*-decalin compounds, whereas both **3** and **4** inhibited DmNobo, but only **3** inhibited mitochondria.

## Discussion

Decalin skeletons are formed via an intramolecular Diels–Alder reaction from linear polyene intermediates biosynthesized by PKS or the PKS module of the PKS-NRPS hybrid enzyme [2, 5]. Four types of decalin with distinct configurations were formed. Decalin-containing tetramic acids isolated from the filamentous fungi were divided into four groups based on their decalin configuration (S1 Fig) [2]. A single biosynthetic pathway for decalin-containing tetramic acid involving the Fsa2-family decalin synthase yielded products with a single decalin configuration. As mentioned above, this group of compounds exhibits structural diversity arising from the methylation pattern and length of the polyketide intermediate, and an amino acid becomes part of the tetramic acid moiety. Thus, evaluating the relationship between decalin

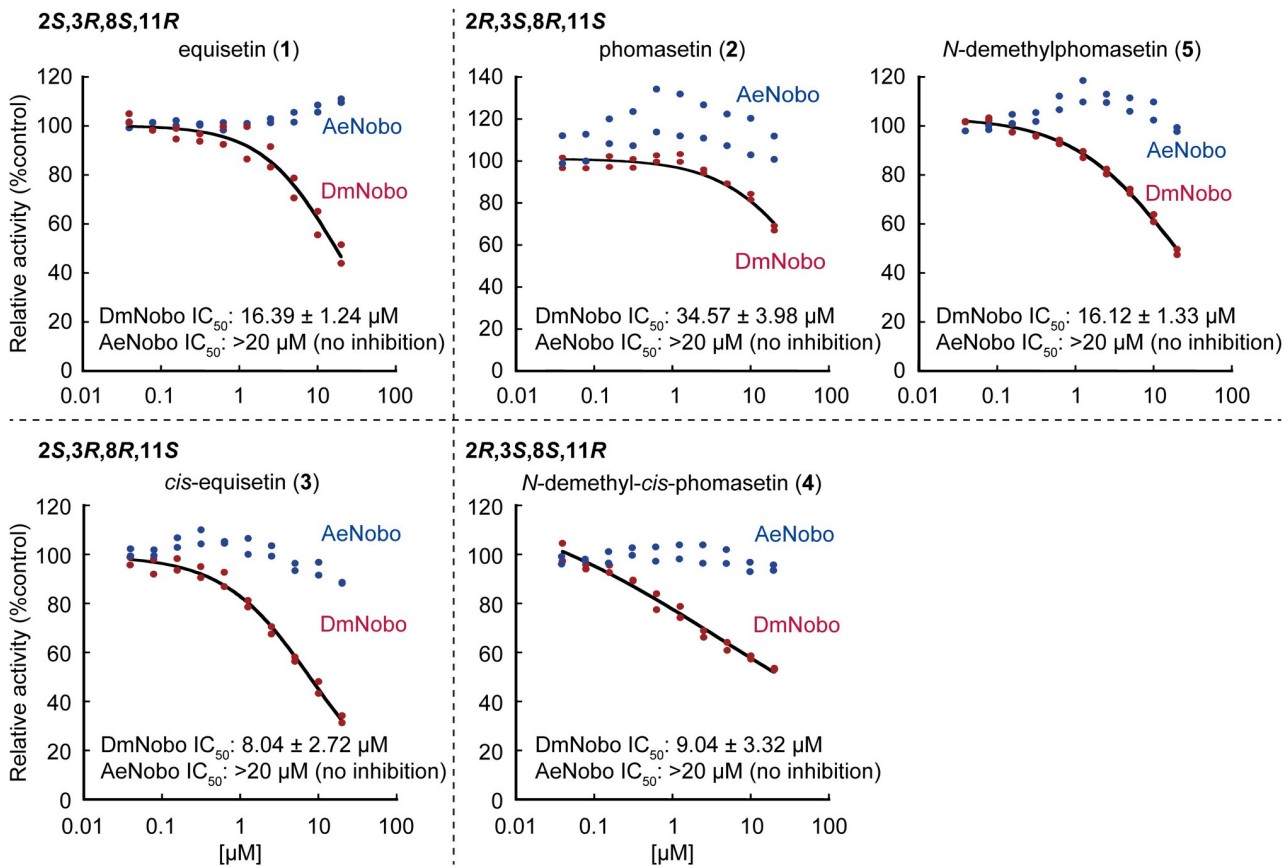

**Fig 3. Decalin-dependent inhibitions of GSH conjugation activities of DmNobo and AeNobo.** Inhibition of the GSH conjugation activities of wild-type DmNobo (red dots) and AeNobo (blue dots) using 3,4-DNADCF in the presence of **1–5**. Each relative activity was defined as the ratio of activity compared between the respective proteins without the compounds. All the data points in duplicate assays are indicated.

configuration and biological activity using these naturally occurring decalin compounds is problematic. Decalin synthase gene knockout resulted in the production of decalin-containing tetramic acids with mixed stereochemistry [6, 7, 9, 13]. Thus, utilizing the genetically engineered mutants derived from the producer fungi of **1** and the enantiomerically opposite **2**, we efficiently collected decalin-containing tetramic acids with all four possible decalin configurations and small differences in substituents (Fig 1).

Decalin-containing tetramic acids have been reported to exhibit a range of biological activities, including antibacterial, antiviral, antifungal, and antitumor activities [1, 2]. Evaluation of biological activities such as cytotoxicity against cancer cell lines, antimicrobial activity, and mitochondrial inhibition demonstrated that the activity for each assay varies depending on the decalin configurations (Fig 4). The *trans* decalin configurations are crucial for exerting strong antibacterial activity against *S. aureus* and some antifungal activities (Table 1). Compounds **1** and **2** and their related compounds have been reported to exhibit antibacterial activity against Gram-positive bacteria [2]. The molecular target of **1** and pyrrolocin C, a related compound with the same decalin configuration as **2** [9], has been proposed to be acetyl-CoA carboxylase [28]. Although the configuration of pyrrolocin C is opposite to that of **1** (i.e., the same as that of **2**), both compounds possess anti-*Staphylococcus* activity and would share the same molecular targets. Moreover, as was the case for **3** and **4**, the *cis*-decalin isomer of pyrrolocin C had

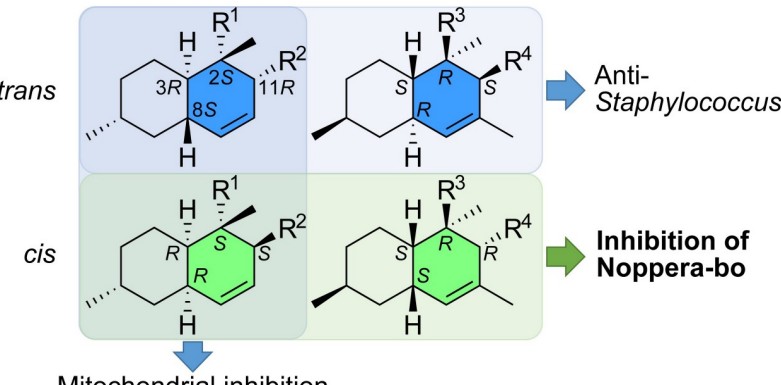

**Fig 4. Configuration-activity relationship of decalin-containing tetramic acids.** Compounds having *trans*-decalin and cis-decalin showed strong anti-Staphylococcus activity and Nobo inhibitory activity, respectively, while the decalins with 3*R* configuration inhibited mitochondrial function.

lower antibacterial activity [9, 28]. Thus, no significant difference in antibacterial activity between the two *trans*-decalin compounds with opposite configurations (2*S*,3*R*,8*S*,11*R* and 2*R*,3*S*,8*R*,11*S*) was present, whereas a change in the configuration from *trans* to *cis* caused a decrease in antibacterial activity. Additionally, compounds **1** and **2** have been reported to inhibit HIV-1 integrase activity approximately equally [4]. In contrast, higher mitochondrial inhibitory activity was observed when *cis*-decalin **3** was used compared to **1**. Unlike its anti-*Staphylococcus* and HIV-1 integrase inhibitory activities, *cis*-decalin with the 2*S*,3*R*,8*R*,11*S* configuration, but not the 2*R*,3*S*,8*S*,11*R*, showed stronger inhibitory activity (Fig 4). The flat shape of both *trans*-decalins acts similarly to their molecular targets, whereas the bent shape of the two *cis*-decalins can be distinguished by the targets. Therefore, engineering decalin configurations is a useful approach for identifying derivatives with better biological activities.

Ecdysteroids are the principal insect steroid hormones and play pivotal roles in regulating many aspects of development and physiology in insects [29–31]. Because ecdysteroids are not biosynthesized in mammals, it has been long thought that the ecdysteroid biosynthesis could be a good target for developing new pesticides that specifically inhibit insect life cycles, with no adverse effects on other animals [32]. The insect steroidogenic GST, Nobo, is a key enzyme in the ecdysteroid biosynthetic pathway [33–35]. Thus, its inhibitors could potentially be used as insecticides to prevent ecdysone-dependent molting and metamorphosis [32]. Previous high-throughput screens to identify and characterize Nobo inhibitors have been conducted [21, 22, 36]. Among them, 17β-estradiol has been identified as a DmNobo inhibitor, which prompted us to examine the inhibitory activity of the decalin compounds against Nobo. Besides 17β-estradiol, flavonoids and dimedone compounds showed a potent Nobo inhibitory activity with $IC_{50}$ of approximately 1 μM. Those inhibitors interact importantly with the conserved Asp/Glu113 of Nobo, forming a strong hydrogen bond with the hydroxyl/carbonyl groups of the compounds [21, 22]. The inhibitory activity of *cis*-decalins **3** and **4** against Nobo was not as high as those of the previously reported compounds, possibly due to the steric hindrance of the compounds. The skeleton of decalin compounds differs significantly from those of the existing Nobo inhibitors. Future studies are needed to elucidate the complex structure of decalin compounds and Nobo by X-ray crystallography and to identify the interacting amino acid residues, leading to the develop of new Nobo inhibitors as insect growth regulators.

## Supporting information

**S1 Table. $^1$H and $^{13}$C NMR chemical shifts of the dimethylester of 8.**
(XLSX)

**S1 Fig. Structural diversity of fungal decalin-containing tetramic acids.** Decalin-containing tetramic acids were divided into four groups based on the configuration of decalin, which is formed from linear polyene intermediates via an intramolecular Diels–Alder reaction by decalin synthase (DS). Compounds and their corresponding DSs identified and those whose DS or biosynthetic gene cluster were not identified are shown in the top and bottom panels, respectively.
(TIF)

**S2 Fig. $^1$H NMR spectrum of dimethylester of 8 in CDCl$_3$.**
(TIF)

**S3 Fig. $^{13}$C NMR spectrum of dimethylester of 8 in CDCl$_3$.**
(TIF)

**S4 Fig. HSQC spectrum of dimethylester of 8 in CDCl$_3$.**
(TIF)

**S5 Fig. $^{13}$C DEPT135 experiment of dimethylester of 8 in CDCl$_3$.**
(TIF)

**S6 Fig. DQF-COSY spectrum of dimethylester of 8 in CDCl$_3$.**
(TIF)

**S7 Fig. HSQC-TOCSY spectrum of dimethylester of 8 in CDCl$_3$.**
(TIF)

**S8 Fig. HMBC spectrum of dimethylester of 8 in CDCl$_3$.**
(TIF)

**S9 Fig. NOESY spectrum of dimethylester of 8 in CDCl$_3$.**
(TIF)

**S10 Fig. Experimental and calculated ECD spectra of the dimethylester of 8.**
(TIF)

**S11 Fig. Structures of 8 and its dimethylester (A) and key 2D NMR correlations of the dimethylester of 8 (B and C).**
(TIF)

**S12 Fig. Optimized conformers of the (2$R$,3$S$,6$S$,8$S$,11$R$,5$'R$)-form of the dimethylester of 8 for calculation of the ECD spectrum with the Boltzmann average (total, 90.0%).**
(TIF)

**S13 Fig. Optimized conformers of the (2$S$,3$R$,6$R$,8$R$,11$S$,5$'S$)-form of the dimethylester of 8 for calculation of the ECD spectrum with the Boltzmann average (total, 100.0%).**
(TIF)

**S14 Fig. Effects of decalin compounds 1–5 and 8 on metabolic phenotypes.** Real-time measurements of the OCR and ECAR in HeLa cells were performed after treating the cells with different concentrations of the test samples. To perform the Seahorse XF Cell Mito Stress Test, the cells were treated with oligomycin A (OMA, 1 μM), FCCP (0.125 μM), and rotenone/antimycin A (R/A, 1 μM each) at the indicated times. Data are mean ± s.d. (n = 3 technical

replicates) from one representative experiment out of three independent experiments.
(TIF)

**S15 Fig. Effects of mevastatin on Noppera-bo (Nobo) activity.** Inhibition of GSH conjugation activities of wild-type DmNobo (red dots) and AeNobo (blue dots) using 3,4-DNADCF was measured in the presence of mevastatin. Relative activity was defined as the ratio of activity between the respective proteins without the compounds.
(TIF)

## Acknowledgments

We acknowledge the computational resources provided by the RIKEN Advanced Center for Computing and Communication (HOKUSAI GreatWave and Big Waterfall).

## Author Contributions

**Conceptualization:** Naoki Kato, Kazuhiko Matsuda, Ryusuke Niwa.

**Funding acquisition:** Naoki Kato, Kazuhiko Matsuda, Hiroyuki Osada, Ryusuke Niwa, Shunji Takahashi.

**Investigation:** Naoki Kato, Kana Ebihara, Toshihiko Nogawa, Yushi Futamura, Kazue Inaba, Akiko Okano, Harumi Aono.

**Methodology:** Ryusuke Niwa.

**Project administration:** Naoki Kato, Hiroyuki Osada, Shunji Takahashi.

**Resources:** Yuuta Fujikawa, Hideshi Inoue.

**Supervision:** Kazuhiko Matsuda, Hiroyuki Osada, Ryusuke Niwa, Shunji Takahashi.

**Validation:** Yushi Futamura.

**Writing – original draft:** Naoki Kato, Toshihiko Nogawa, Yushi Futamura, Ryusuke Niwa.

**Writing – review & editing:** Naoki Kato, Kana Ebihara, Yushi Futamura, Kazuhiko Matsuda, Ryusuke Niwa.

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
