## [Decision Letter · Decision Letter 0]

4 Jul 2023

PONE-D-23-17309cis-Decalin-containing tetramic acids as inhibitors of insect steroidogenic glutathione S-transferase Noppera-boPLOS ONE

Dear Dr. Kato,

Thank you for submitting your manuscript to PLOS ONE. After careful consideration, we feel that it has merit but does not fully meet PLOS ONE’s publication criteria as it currently stands. Therefore, we invite you to submit a revised version of the manuscript that addresses the points raised during the review process. Please submit your revised manuscript by Aug 18 2023 11:59PM. If you will need more time than this to complete your revisions, please reply to this message or contact the journal office at plosone@plos.org. Please include the following items when submitting your revised manuscript:A rebuttal letter that responds to each point raised by the academic editor and reviewer(s). You should upload this letter as a separate file labeled 'Response to Reviewers'.A marked-up copy of your manuscript that highlights changes made to the original version. You should upload this as a separate file labeled 'Revised Manuscript with Track Changes'.An unmarked version of your revised paper without tracked changes. You should upload this as a separate file labeled 'Manuscript'.

We look forward to receiving your revised manuscript.

Kind regards,

A Ganesan

Academic Editor

PLOS ONE

Journal Requirements:

2. We note that this submission includes NMR spectroscopy data. We would recommend that you include the following information in your methods section or as Supporting Information files:

1) The make/source of the NMR instrument used in your study, as well as the magnetic field strength. For each individual experiment, please also list: the nucleus being measured; the sample concentration; the solvent in which the sample is dissolved and if solvent signal suppression was used; the reference standard and the temperature.

2) A list of the chemical shifts for all compounds characterised by NMR spectroscopy, specifying, where relevant: the chemical shift (δ), the multiplicity and the coupling constants (in Hz), for the appropriate nuclei used for assignment.

3)The full integrated NMR spectrum, clearly labelled with the compound name and chemical structure.

We also strongly encourage authors to provide primary NMR data files, in particular for new compounds which have not been characterised in the existing literature. Authors should provide the acquisition data, FID files and processing parameters for each experiment, clearly labelled with the compound name and identifier, as well as a structure file for each provided dataset. See our list of recommended repositories here: https://journals.plos.org/plosone/s/recommended-repositories

3. Please expand the acronym “JSPS” (as indicated in your financial disclosure) so that it states the name of your funders in full.

"This work was supported by JSPS KAKENHI (Grant numbers 18K19163 to RN, 19H04665 and 20K05872 to NK, and 21H04718 to RN, NK, and KM) and the Support for Pioneering Research Initiated by the Next Generation, Japan Science and Technology Agency (JST SPRING; Grant number JPMJSP2124). KE received a fellowship from JST SPRING. "

Reviewers' comments:

Reviewer's Responses to Questions

**Comments to the Author**

1. Is the manuscript technically sound, and do the data support the conclusions?

Reviewer #1: Yes

Reviewer #2: Yes

2. Has the statistical analysis been performed appropriately and rigorously? 

Reviewer #1: Yes

Reviewer #2: Yes

3. Have the authors made all data underlying the findings in their manuscript fully available?

Reviewer #1: No

Reviewer #2: Yes

4. Is the manuscript presented in an intelligible fashion and written in standard English?

Reviewer #1: Yes

Reviewer #2: Yes

5. Review Comments to the Author

Reviewer #1: The manuscript entitled "cis-Decalin-containing tetramic acids as inhibitors of insect steroidogenic glutathione S-transferase Noppera-bo," is a resrach article which focuses on the biological evaluation of 8 Decalin-containing tetramic acid derivatives obtained from filamentous fungi and their genetically modified mutants.

The authors have presented a primary structure-activity relationship analysis, particularly regarding decalin configurations and various investigated biological activities such as cytotoxicity, antimicrobial activity, mitochondrial inhibition, and inhibition of the insect steroidogenic glutathione S-transferase Noppera-bo.

Overall, the scientific conceptualisation of the study is sound, and the methodologies employed are well-designed. I recommend publication of this manuscript with a minor revision.

Specifically, I would like to invite the authors to include representative graphs or chemical schemes illustrating the structure-activity relationships for each investigated biological activity (Figuring out which group/decalin configuration is crucial and relevant to biological activity).

This addition would enhance the informativeness of the manuscript and facilitate the narrative flow of this section.

Reviewer #2: Kato et al describe the report of the biological activities of cis-decalin-containing tetramic acids. The experimental data obtained were accurate and support the conclusions. The text is written in clear English, which helps readers understand the contents, although some parts of the text seem a bit lacking in explanation. The following are my suggestions for revisions.

1. P2L12. Has the relationship between the configuration and biological activity of decalin been demonstrated through this paper? It would be correct to say that the activity for each assay varies depending on the decalin configurations.

2. P3L26. The readers may be interested to know what type of fungus was used in this study. Why not describe it properly at least once in the introduction?

3. P4L12. It should be clearly stated that it was corrected for the chloroform signal.

4. P5L12-P7L1. Please add S figures showing the correlations of 2D NMR (COSY, TOCSY, HMBC, NOESY) used in the structure determination of each compound.

5. P8L21. I am not sure what you mean by the use of parentheses in the notation.

6. P10L12. bacteria -> bacterium

7. P10L23-P11-L7. The circumstances that led to the decision to examine mitochondrial function here are too abrupt and unclear. Please describe in more detail two points: why you decided to do this test and how you will evaluate it in this study.

8. P11L21. The authors describe that they looked up the structure of cis-decalin and ecdysteroid because they are similar, but there is nothing in the paper to determine how similar they actually are. This is a very important part, so please move on to the explanation of the results with logical guidance.

9. P11L23-L28. From this description alone, it is not clear what principle the assay is based on. Please add more explanation.

10. P13L27-P14L7. The title suggests that the authors consider this part of the results to be the most important. However, the DISCUSSION is basically just a repetition of the results, and nothing useful to the readers is discussed, such as how to interpret the results obtained, comparison of activity with existing inhibitors, or usefulness as an inhibitor.

6. PLOS authors have the option to publish the peer review history of their article (what does this mean?). If published, this will include your full peer review and any attached files.

Reviewer #1: No

Reviewer #2: No

---

## [Author Response · Author response to Decision Letter 0]

15 Aug 2023

Response to the editor

->We have confirmed that our manuscript meets the style requirements, according to the guidelines attached. 

2. We note that this submission includes NMR spectroscopy data. We would recommend that you include the following information in your methods section or as Supporting Information files:

1) The make/source of the NMR instrument used in your study, as well as the magnetic field strength. For each individual experiment, please also list: the nucleus being measured; the sample concentration; the solvent in which the sample is dissolved and if solvent signal suppression was used; the reference standard and the temperature.

->Information requested, including the NMR instrument, magnetic field strength, and solvent, are listed in the Materials and methods section (p. 4, lines 17-20). 

2) A list of the chemical shifts for all compounds characterised by NMR spectroscopy, specifying, where relevant: the chemical shift (δ), the multiplicity and the coupling constants (in Hz), for the appropriate nuclei used for assignment.

->Compound 8 is a new compound reported in this study, and a list of chemical shifts of this compound is included in the Supporting Information as S1 Table.

3)The full integrated NMR spectrum, clearly labelled with the compound name and chemical structure.

->We have corrected S2 Fig for 1H NMR spectrum of dimethylester of 8 as suggested.

3. Please expand the acronym “JSPS” (as indicated in your financial disclosure) so that it states the name of your funders in full.

->JSPS stands for Japan Society for the Promotion of Science. Please revise it on the online submission form.

"This work was supported by JSPS KAKENHI (Grant numbers 18K19163 to RN, 19H04665 and 20K05872 to NK, and 21H04718 to RN, NK, and KM) and the Support for Pioneering Research Initiated by the Next Generation, Japan Science and Technology Agency (JST SPRING; Grant number JPMJSP2124). KE received a fellowship from JST SPRING. "

->We declare that the funders had no role in study design, data collection and analysis, decision to publish, or preparation of the manuscript. Please revise it on the online submission form.

Response to Reviewer 1

I would like to invite the authors to include representative graphs or chemical schemes illustrating the structure-activity relationships for each investigated biological activity (Figuring out which group/decalin configuration is crucial and relevant to biological activity).

This addition would enhance the informativeness of the manuscript and facilitate the narrative flow of this section.

->Thank you very much for this comment. We have added a figure showing a relationship between decalin configuration and biological activities as Fig 4 (p. 13, lines 27-28, p.14, lines 15, and p14, lines 20-23). We hope that it would help readers understand which configurations are important for anti-Staphylococcus, mitochondrial inhibition, and Nobo inhibition.

Response to Reviewer 2

1. P2L12. Has the relationship between the configuration and biological activity of decalin been demonstrated through this paper? It would be correct to say that the activity for each assay varies depending on the decalin configurations.

->According to this suggestion, we have revised the sentence as follows: “Our evaluation of …activities, demonstrated that the activity for each assay varies depending on the decalin configurations.” Along with this change, we have also modified a sentence in the second paragraph of the Discussion section (p. 13, lines 27, 28).

2. P3L26. The readers may be interested to know what type of fungus was used in this study. Why not describe it properly at least once in the introduction?

->We have inserted the information on the producer fungi and their genetically engineered mutants used in this study into the last paragraph of the Introduction section.

3. P4L12. It should be clearly stated that it was corrected for the chloroform signal.

->Corrected as suggested.

4. P5L12-P7L1. Please add S figures showing the correlations of 2D NMR (COSY, TOCSY, HMBC, NOESY) used in the structure determination of each compound.

->S11 Fig shows chemical structures of compound 8, which was isolated in this study, and its dimethyl ester derivative. It also includes key correlations of 2D NMR used in the structure determination (p. 19, lines 18, 19).

5. P8L21. I am not sure what you mean by the use of parentheses in the notation.

->According to this suggestion, we have removed the parentheses and modified the sentence to make clear which is opposite to which (p. 9, lines 2, 3).

6. P10L12. bacteria -> bacterium

->Corrected as suggested.

7. P10L23-P11-L7. The circumstances that led to the decision to examine mitochondrial function here are too abrupt and unclear. Please describe in more detail two points: why you decided to do this test and how you will evaluate it in this study.

->Konig et al. have reported that equisetin (1) inhibit mitochondrial respiratory chain and subsequent oxidative phosphorylation indirectly by influencing substrate anion carriers of mitochondria in eukaryotic cells (ref25). This is a reason why we have decided to investigate the mitochondrial inhibitory activity of 1 and its analogs using flux analyzer. The flux analyzer can measure the mitochondrial respiratory activity, and demonstrated that 1 inhibited mitochondrial respiratory chain as previously reported. Then, we evaluated inhibitory activity of the decalin compounds 2-5, 8. Among the compound tested, 1 and its cis-decalin analog 3 showed the inhibitory activity, suggesting importance of the 3R configuration in exerting the activity. According to this comment, we have modified the paragraph describing mitochondrial inhibitory activity of decalin compounds by flux analyzer (p.11, lines 8-20) and the Fig 2 legend (p.11, lines 25-27).

8. P11L21. The authors describe that they looked up the structure of cis-decalin and ecdysteroid because they are similar, but there is nothing in the paper to determine how similar they actually are. This is a very important part, so please move on to the explanation of the results with logical guidance.

->The phrase pointed has been rephrased as follows: “We postulated that the decalin-containing tetramic acids may have activity against insect steroidogenesis, in addition to the aforementioned biological activities, according to the following reasons. (i) Carbon skeleton of cis-decalin is structurally similar to that of the AB-ring moiety of ecdysteroids. (ii) Lovastatin and mevastatin with a decalin skeleton have been reported to inhibit the HMG-CoA reductase, a key enzyme in steroid biosynthesis (ref26, 27). (iii) the mammalian female hormone 17β-estradiol inhibits Noppera-bo (Nobo), a GST involved in insect steroidogenesis (ref22). Hence, to test this hypothesis, we investigated the inhibitory activity of this group of compounds against Nobo.” Corresponding to this change, the following sentences have also been revised (p.12, lines 8-16).

9. P11L23-L28. From this description alone, it is not clear what principle the assay is based on. Please add more explanation.

->According to this suggestion, we have added a sentence to explain how the in vitro enzyme assay system works (p.12, lines 19, 20). Along with this change, we have revised the positions of two abbreviations, glutathione S-transferase (GST) and glutathione (GSH) throughout the manuscript (p. 4, line 5, and p. 13, line 2).

10. P13L27-P14L7. The title suggests that the authors consider this part of the results to be the most important. However, the DISCUSSION is basically just a repetition of the results, and nothing useful to the readers is discussed, such as how to interpret the results obtained, comparison of activity with existing inhibitors, or usefulness as an inhibitor.

->As the reviewer pointed out, we believe that one of the highlights of the manuscript is finding Nobo inhibitory activity of the decalin compounds. According to the reviewer’s comment, we have thoroughly modified the third paragraph of the Discussion section to discuss the merit of this study. The revised paragraph includes an introduction of ecdysteroids and Nobo, a comparison with inhibitors previously reported, and an interpretation of the results and prospects (p.14, line 25-p. 15, line 14). 

We would like to thank the reviewer for his/her comments. Our revised manuscript has been greatly improved, and we hope that it will cross the threshold for the publication

---

## [Editor Report · Decision Letter 1]

17 Aug 2023

cis-Decalin-containing tetramic acids as inhibitors of insect steroidogenic glutathione S-transferase Noppera-bo

PONE-D-23-17309R1

Dear Dr. Kato,

We’re pleased to inform you that your manuscript has been judged scientifically suitable for publication and will be formally accepted for publication once it meets all outstanding technical requirements.

Kind regards,

A Ganesan

Academic Editor

PLOS ONE
---

## [Editor Report · Acceptance letter]

22 Aug 2023

PONE-D-23-17309R1 

*cis*-Decalin-containing tetramic acids as inhibitors of insect steroidogenic glutathione *S*-transferase Noppera-bo 

Dear Dr. Kato:

I'm pleased to inform you that your manuscript has been deemed suitable for publication in PLOS ONE. Congratulations! Your manuscript is now with our production department. 

Kind regards, 

on behalf of

Prof. A Ganesan 

Academic Editor

PLOS ONE